# ANTI-EXPOSURE BIAS IN DIFFUSION MODELS

**Junyu Zhang**
School of Computer Science and Engineering
Central South University

**Daochang Liu**
School of Physics, Mathematics and Computing
University of Western Australia

**Eunbyung Park**
Department of Artificial Intelligence, Yonsei University

**Shichao Zhang**
School of Computer Science and Engineering
Guangxi Normal University

**Chang Xu**[*]
School of Computer Science
University of Sydney

## ABSTRACT

Diffusion models (DMs) have achieved record-breaking performance in image generation tasks. Nevertheless, in practice, the training-sampling discrepancy, caused by score estimation error and discretization error, limits the modeling ability of DMs, a phenomenon known as exposure bias. To alleviate such exposure bias and further improve the generative performance, we put forward a prompt learning framework built upon a lightweight prompt prediction model. Concretely, our model predicts an anti-bias prompt for the generated sample at each sampling step, aiming to compensate for the exposure bias that arises. Following this design philosophy, our framework rectifies the sampling trajectory to match the training trajectory, thereby reducing the divergence between the target data distribution and the modeling distribution. To train the prompt prediction model, we simulate exposure bias by constructing training data and introduce a time-dependent weighting function for optimization. Empirical results on various DMs demonstrate the superiority of our prompt learning framework across three benchmark datasets. Importantly, the optimized prompt prediction model effectively improves image quality with only a 5% increase in sampling overhead, which remains negligible. Our code is available at: Anti_Exposure_Bias.

## 1 INTRODUCTION

Diffusion models (DMs) (Sohl-Dickstein et al., 2015; Song & Ermon, 2019; Ho et al., 2020) represent a novel generative paradigm that has become *de facto* standard for image generation, and also showcasing impressive results in many downstream tasks (Luo et al., 2023; Shue et al., 2023; Liu et al., 2023a; Mokady et al., 2023). In particular, the seminal work (Song et al., 2021b) unifies the design philosophy of DMs through continuous diffusion using stochastic differential equations (SDEs), boosting them for achieving start-of-the-art image quality (Kim et al., 2023a; Peebles & Xie, 2023) and improved mode coverage (Kingma et al., 2021; Song et al., 2021a; Lu et al., 2022a; Kim et al., 2022). More recently, stable diffusion (Rombach et al., 2022) has bridged the gap in both text-to-image (Nichol et al., 2022; Ramesh et al., 2022; Saharia et al., 2022) and text-to-video generation (Blattmann et al., 2023; Khachatryan et al., 2023), further enhancing the modeling capability for high fidelity, controllable content synthesis (Gao et al., 2023; Ruiz et al., 2023) and demonstrating great potential for practical applications (Xu et al., 2024; Sauer et al., 2023).

The core idea of DMs is to establish a diffusion path between the target data distribution and a prior distribution and simulate this path in the opposite direction for image generation, dubbed forward diffusion and reverse sampling, respectively (Sohl-Dickstein et al., 2015; Song & Ermon, 2020). In the diffusion process, a forward SDE is employed to formulate the diffusion path via perturbing the data distribution with a well-designed multilevel noise schedule (Song et al., 2021b; Karras et al.,

---

[*]Corresponding Author

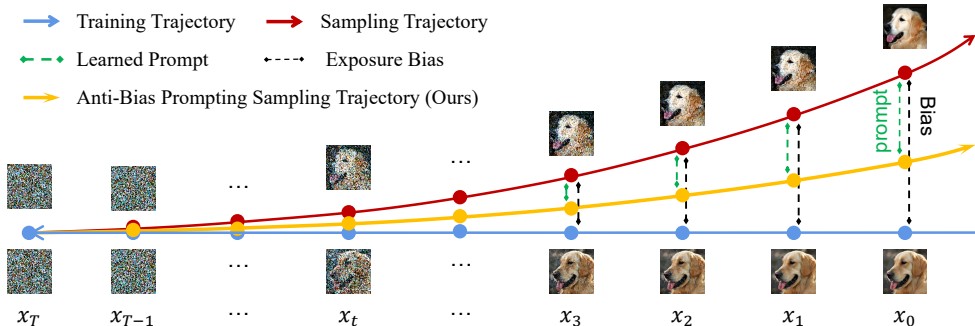

Figure 1: **Anti-Exposure Bias Prompt Learning.** The exposure bias arises from the training-sampling discrepancy of model inputs. As clearly illustrated in this figure, the sampling trajectory gradually deviates from the training trajectory due to accumulated score estimation and discretization errors at each time step, leading to an increase in bias. To alleviate this bias and enhance modeling performance, our prompt learning framework employs a lightweight parameterized model to generate an anti-bias prompt from the current time-step sample, compensating for exposure bias in the model input at the next time step. In addition to delivering excellent performance, the resulting extra sampling overhead is less than 5%, which can be considered negligible (Best viewed in color).

2022). Concretely, different noise scales represent distinct time steps in the diffusion path, with the transition between adjacent time steps characterized by a Gaussian transition kernel. The reverse sampling can be achieved by traversing the diffusion path with the opposite direction (Kim & Ye, 2022; Lu et al., 2022b; Zhang & Chen, 2022). Crucially, the reverse process satisfies a reverse-time SDE or a probability flow (PF) ordinary differential equation (ODE) (Song et al., 2021b). Both can be derived from the forward SDE by considering the score of the marginal probability densities as a function of time (Anderson, 1982; Luo, 2022). We can, therefore, approximate the reverse-time S/ODE by training a time-dependent deep neural network to estimate the scores (Hyvärinen & Dayan, 2005; Song et al., 2020b; Song & Ermon, 2019), and generate new images using numerical S/ODE solvers (Song et al., 2021b; Lu et al., 2022b; Zhang & Chen, 2022; Zhang et al., 2023).

However, in practice, a training-sampling discrepancy exists at each time step concerning the input to the time-dependent neural network model, resulting in an exposure bias problem (Ning et al., 2023b; Li et al., 2023a; Ning et al., 2023a) and, consequently, a degradation in image quality (Kim et al., 2023a). As illustrated in Figure 1, this issue arises because, during training, the model inputs for DMs are derived from ground truth samples, while during sampling, the inputs are predictions from the previous time step. Practically, the predictions cannot completely match the ground truth samples due to two fundamental errors: the score estimation error (Bao et al., 2022a) and the discretization error (Zhang & Chen, 2022). The score estimation error is primarily caused by the score conflict (Hang et al., 2023), data sparsity (Kim et al., 2022) and model capacity (Karras et al., 2022), as well as an imperfect diffusion schedule (Dhariwal & Nichol, 2021). Regarding the latter, since integration in high-dimensional spaces is intractable, we can only approximate reverse-time S/ODEs using numerical solvers to the best of our ability, which inevitably results in discretization error (Wang et al., 2021; Bao et al., 2022c; Lu et al., 2022b). Due to these two types of errors, exposure bias inherently arises during the sampling process in DMs. Furthermore, this bias becomes increasingly pronounced along the sampling trajectory, as each time step accumulates newly resulting score estimation and discretization errors (De Bortoli et al., 2021; Xiao et al., 2021). As a result, exposure bias has a significant impact on the generative performance of DMs.

To illuminate exposure bias, we first thoroughly examine the training-sampling discrepancy problem from an analytical perspective. Theoretically, the denoising distribution between adjacent time steps follows a naive Gaussian distribution (Ho et al., 2020; Sohl-Dickstein et al., 2015). We can, therefore, approximate it by a Gaussian transition kernel. However, when considering the phenomenon of exposure bias, there is a gap at each time step between the true Gaussian distribution and the transition kernel modeled by a pre-trained DM (Luo et al., 2024). Moreover, the magnitude of this gap increases when fewer sampling time steps are used (Kim et al., 2023a), resulting in a reduction in image quality, as the Gaussian assumption holds only in the infinitesimal limit of small denoising steps (Xiao et al., 2021). While the gap in each denoising step can be quantified using the Kullback-

Leibler (KL) divergence between the true Gaussian kernel and the modeling transition kernel at the current time step, accessing these latent distributions is not feasible (Kingma & Gao, 2024).

To remedy this, we propose a novel exposure bias prompt learning framework that uses a parameterized prompt prediction model to rectify biases in the generated samples at each time step. To effectively optimize the prompt prediction model, we construct training data that simulates exposure bias and introduce a time-dependent weighting function for stable training. During the sampling process, the optimized prompt prediction model predicts an exposure bias prompt based on the generated sample at the current time step. This prompt is then used to correct the bias in the model input for the next step. The combination of the generated sample and its anti-exposure bias prompt creates an improved sample input for the subsequent time step. In this manner, we alleviate the exposure bias caused by the score estimation and discretization errors during sampling, effectively rectifying sampling trajectories through iterative execution of this process. Importantly, the prompt prediction model is a lightweight backbone that requires only a 5% increase in sampling time, which can be considered negligible. Furthermore, our framework provides significant flexibility, enabling enhancements to guidance sampling (Dhariwal & Nichol, 2021), alleviating exposure bias in latent diffusion (Vahdat et al., 2021; Rombach et al., 2022), and improving fast samplers (Song et al., 2020a; Bao et al., 2022b; Zhang & Chen, 2022; Lu et al., 2022b). Notably, compared to merely increasing DM parameters to alleviate exposure bias, our framework can serve as a plug-in to enhance various DMs parameterized by different model sizes and architectures. The former requires training the model from scratch, incurring significant computational costs and human effort, while we only necessitate training a lightweight backbone, demonstrating both flexibility and efficiency.

In a nutshell, our contributions can be summarized as follows: 1) We analyze the phenomenon of exposure bias in DMs caused by score estimation and discretization errors; 2) To alleviate exposure bias and enhance generative performance, we propose a prompt learning framework that employs a lightweight parameterized model to predict an anti-bias prompt for rectifying the next model input; 3) A novel training strategy is proposed to simulate exposure bias and ensure stable training; 4) Extensive experiments demonstrate the effectiveness of our prompt learning framework across various datasets and different DMs, with only a negligible increase in sampling overhead.

## 2 BACKGROUND

**Overview** DMs (Song & Ermon, 2019; Ho et al., 2020) are a new class of generative models that synthesize images by gradually denoising random points sampled from a prior distribution. Specifically, for a given $D$-dimensional image $x_0$, we assume it satisfies a distribution $x_0 \sim p(x_0)$. Thus, the diffusion path leading to a prior distribution can be constructed via the following forward SDE:

$$dx = \boldsymbol{F}_t x dt + \boldsymbol{G}_t d\omega, \tag{1}$$

where $\boldsymbol{F}_t \in \mathbb{R}^{D \times D}$ denotes the linear drift coefficient, $\boldsymbol{G}_t \in \mathbb{R}^{D \times D}$ denotes the diffusion coefficient, $\omega$ is a standard Wiener process and $t \sim U[0, 1]$. Under some mild assumptions (Song et al., 2021b), the forward SDE in Eq. (1) is associated with a reverse-time diffusion process:

$$dx = \left[ \boldsymbol{F}_t x - \boldsymbol{G}_t \boldsymbol{G}_t^T \nabla \log p_t(x) \right] dt + \boldsymbol{G}_t d\bar{\omega}, \tag{2}$$

where $\bar{\omega}$ denotes a standard Wiener process in the reverse-time direction, and $\nabla \log p_t(x)$ represents the gradient of the log probability density with respect to the perturbed data at time step $t$, a.k.a. score (Hyvärinen & Dayan, 2005; Vincent, 2011). In theory, with a known prior distribution $\pi$, such as the normal distribution, one can generate new images via solving Eq. (2) using initial samples $x_T \sim \pi$ (Anderson, 1982).

**Training** In practice, $\nabla \log p_t(x)$ is inaccessible due to the high dimensionality of data, which leads to the analytical intractability of the probability density function (Hyvärinen & Dayan, 2005). To remedy this, prior works (Song et al., 2020b; Vincent, 2011; Song & Ermon, 2019) employ a time-dependent neural network $s_\theta(x_t, t)$ to approximate the score:

$$\mathcal{J}_{\text{SM}}\left(\theta; \omega(\cdot)\right) = \frac{1}{2} \int_0^1 \mathbb{E}_{x_0, x_t} \left[ \omega(t) \left\| \nabla \log p_{0t}(x_t | x_0) - s_\theta(x_t, t) \right\|_2^2 \right] dt. \tag{3}$$

Here, $\nabla \log p_{0t}(x_t | x_0)$ has a closed form expression as $p_{0t}(x_t | x_0)$ is a simple Gaussian distribution obtained from a given SDE (Song et al., 2021b), and $\omega(t)$ denotes a time-dependent weighting

function used for stable training (Kingma et al., 2021; Kim et al., 2022). When implementing advanced score matching techniques, Eq. (3) can be optimized using empirical samples via Monte Carlo methods (Hyvärinen & Dayan, 2005; Song & Ermon, 2019; 2020).

## 3 DISCUSSION

### 3.1 EXPOSURE BIAS PHENOMENON

**Score Estimation Error** Once $s_\theta(x_t, t) \approx \nabla \log p_t(x)$ is matched for almost all $x \in \mathbb{R}^D$ and $t \sim U[0, 1]$, one enables to generate images by solving Eq. (2) with $\nabla \log p_t(x)$ replaced by $s_\theta(x_t, t)$:

$$dx = \left[ \boldsymbol{F}_t x - \boldsymbol{G}_t \boldsymbol{G}_t^T s_\theta(x_t, t) \right] dt + \boldsymbol{G}_t d\omega. \tag{4}$$

However, this process will results in score estimation error because of the discrepancy between $\nabla \log p_t(x)$ and $s_\theta(x_t, t)$. This discrepancy is primarily attributed to factors such as data bias, model robustness, and training techniques (Kim et al., 2022; Hoogeboom et al., 2023), which cannot be easily resolved by merely increasing model parameters due to the intrinsic limitations in DMs.

**Discretization Error** In practice, directly solving the integral in Eq. (4) is intractable. Instead, it is approximated by discretizing it into $T$ steps with $T - 1$ intervals, where the transition from time step $t + 1$ to $t$ is governed by a Gaussian kernel $q(x_t \mid x_{t+1})$. For simplicity, we next investigate the discretization error via using the PF ODE, where $x_t$ can be obtained via the following formulation:

$$x_t = \Psi(t, t+1)x_{t+1} + \int_{t+1}^t \Psi(t, \tau) \left[ -\frac{1}{2} \boldsymbol{G}_\tau \boldsymbol{G}_\tau^T s_\theta(x_\tau, \tau) \right] d\tau, \tag{5}$$

where $\frac{\partial \Psi(t, t+1)}{\partial t} = \boldsymbol{F}_t \Psi(t, t+1)$, and $\Psi(t+1, t+1) = \boldsymbol{I}$ represents the transition function from time $t + 1$ to time $t$, which can be derived from $\boldsymbol{F}_\tau$ (Zhang & Chen, 2022). In this manner, new images $x_0$ can be generated by iteratively solving Eq. (5) from the initial time step $T$ to the final time step. Nevertheless, it is also intractable to directly solve the integral part in Eq. (5) because of its ultra-high dimensional nature. In practice, one can utilize a numerical solver (Lu et al., 2022b; Karras et al., 2022; Zhang & Chen, 2022; Li et al., 2023b) to approximate each integral part:

$$\hat{x}_t = \Psi(t, t+1)x_{t+1} + \frac{\Delta t}{2} \boldsymbol{G}_{t+1} \boldsymbol{G}_{t+1}^T s_\theta(x_{t+1}, t+1), \tag{6}$$

where $\Delta t$ is the integration interval between time step $t + 1$ and $t$. For simplicity, we demonstrate only the first-order Euler sampler for solving the integral part. Obviously, using Eq. (6) to solve each integral instead of Eq. (5) will cause the discretization error. This happens because linear solutions provide only a rough approximation of the integral, particularly over large integration intervals.

**Exposure Bias Phenomenon** When accounting for score estimation and discretization errors, exposure bias occurs at each time step along the sampling trajectory, as shown Figure 1. Formally, the model inputs during training are derived from the ground truth images, while the inputs during sampling are the model prediction outputs from previous steps. Due to these two errors, the model predictions cannot exactly match the ground truth value, leading to the exposure bias phenomenon. Based on the analysis, the modeling Gaussian kernel can be formulated as $p_\theta(\hat{x}_t \mid x_{t+1})$. In this context, the exposure bias at each time step is actually the discrepancy between the ground truth output $q(x_t \mid x_{t+1})$ and its predicted output $p_\theta(\hat{x}_t \mid x_{t+1})$. Although we can express it as a KL divergence $D_{\mathrm{KL}}(q(x_t \mid x_{t+1}) \parallel p_\theta(\hat{x}_t \mid x_{t+1}))$, directly minimizing this KL divergence to alleviate exposure bias is intractable, as we do not have access to the true Gaussian kernel.

### 3.2 DESIGN PRINCIPLE

To elucidate exposure bias, we conduct an in-depth investigation into the gap between the true Gaussian kernel $q(x_t \mid x_{t+1})$ and its modeling counterpart $p_\theta(\hat{x}_t \mid x_{t+1})$ from an analytical perspective. Concretely, in each sampling iteration, we can formulate the distinction between the ground truth sample and its biased sample as follows:

$$\Phi(\hat{x}_t, x_t) = \underbrace{\int_{t+1}^t \Psi(t, \tau) \left[ -\frac{1}{2} \boldsymbol{G}_\tau \boldsymbol{G}_\tau^T \nabla \log p(x_\tau) \right] d\tau}_{\text{integral term}} - \underbrace{\frac{\Delta t}{2} \boldsymbol{G}_{t+1} \boldsymbol{G}_{t+1}^T s_\theta(x_{t+1}, t+1)}_{\text{linear term}}. \tag{7}$$

Here, $\Phi(\hat{x}_t, x_t)$ represents the exposure bias in the sample $\hat{x}_t$ compared to $x_t$, stemming from discretization and score estimation errors. Though we do not have access to integral term in Eq. (7), we can employ $\Phi$ to present the ground truth sample as $x_t = \hat{x}_t + \Phi(\hat{x}_t, x_t)$. In this manner, we can present this formulation as a transition kernel $p(x_t \mid \hat{x}_t)$. Formally, we have the expression $q(x_t) = \int p(x_t \mid \hat{x}_t)p(\hat{x}_t)d\hat{x}_t$. However, we cannot directly obtain $p(\hat{x}_t)$ due to its high-dimensional property. In theory, the score of the integral term in Eq. (7) is based on $x_{t+1}$, thereby also forming the basis of $\Phi(\hat{x}_t, x_t)$. Moreover, $\hat{x}_t$ is actually derived from $x_{t+1}$ via Eq. (6). It is reasonable to reformulate the transition kernel $p(x_t \mid \hat{x}_t)$ as $p(x_t \mid \hat{x}_t, x_{t+1})$. We can deconstruct the true Gaussian kernel $q(x_t \mid x_{t+1})$ into the combination of the modeling kernel and an extra transition kernel $p(x_t \mid \hat{x}_t, x_{t+1})$ as follows:

$$q(x_t \mid x_{t+1}) = \int p(x_t \mid \hat{x}_t, x_{t+1})p_\theta(\hat{x}_t \mid x_{t+1})d\hat{x}_t. \tag{8}$$

In Eq. (8), $p_\theta(\hat{x}_t \mid x_{t+1})$ is the modeling Gaussian kernel simulated by a pre-trained DM, which is fixed during sampling. Thus, we can mitigate exposure bias at each time step by using the newly introduced anti-bias kernel $p(x_t \mid \hat{x}_t, x_{t+1})$, contributing to a smaller training-sampling discrepancy.

## 3.3 Convergence Investigation

From a theoretical perspective, our anti-bias transition kernel $p(x_t \mid \hat{x}_t, x_{t+1})$ can help the sampling bound better align with the training KL divergence $D_{\mathrm{KL}}(p(x_0) \parallel p_\theta(\hat{x}_0))$. To illustrate this advantage, we first review the diffusion training objective function, which serves as the theoretical bound for the optimization tasks. Specifically, DMs aim to minimize $D_{\mathrm{KL}}(p(x_0) \parallel p_\theta(\hat{x}_0))$ via optimizing the score matching loss with the weighting function $g(\cdot)^2$ (Song et al., 2021a; Lu et al., 2022a):

$$D_{\mathrm{KL}}(p(x_0) \parallel p_\theta(\hat{x}_0)) \leq D_{\mathrm{KL}}(p(x_T) \parallel \pi) + \mathcal{J}_{\mathrm{SM}}\left(\theta; g(\cdot)^2\right).$$

Here, $g(\cdot)^2$ is the diffusion coefficient in forward SDE, and $\pi$ is a prior distribution. In this context, we can achieve an optimized KL divergence via minimizing $\mathcal{J}_{\mathrm{SM}}\left(\theta; g(\cdot)^2\right)$, as $D_{\mathrm{KL}}(p(x_T) \parallel \pi)$ is a constant. However, during sampling, this bound will be enlarged due to exposure bias phenomenon. Concretely, the score matching in Eq. (3) essentially optimizes the discrepancy between $\nabla \log p_{0t}(x_t|x_0)$ and $s_\theta(x_t, t)$. During training, the input for $s_\theta(x_t, t)$ consists of the training data perturbed by a noise scale at time step $t$, whereas during sampling, the input for $s_\theta(\hat{x}_t, t)$ is the bias sample $\hat{x}_t$. In this context, $s_\theta(\hat{x}_t, t)$ cannot exactly match $\nabla \log p_{0t}(x_t|x_0)$ because of the discrepancy between $\hat{x}_t$ and $x_t$, thus amplifying the upper bound of $D_{\mathrm{KL}}(p(x_0) \parallel p_\theta(\hat{x}_0))$. Ideally, our framework can help $s_\theta(\hat{x}_t, t)$ better approximate $s_\theta(x_t, t)$ since the anti-bias kernel $p(x_t \mid \hat{x}_t, x_{t+1})$ can enable $\hat{x}_t$ to match $x_t$, resulting in a sampling bound closer to the the training bound. By adhering to this design philosophy, our framework enables the simulation of a more realistic transport path between the target data and prior distributions, thereby enhancing generative performance.

## 4 Prompt Learning Framework

To alleviate exposure bias, we propose a novel prompt learning framework that parameterizes a lightweight model $v_\phi(\cdot)$ to simulate the anti-bias kernel $p(x_t \mid \hat{x}_t, x_{t+1})$ in Eq. (8). Concretely, we simulate the bias and introduce a time-dependent weighting function to train the model. Once the model is optimized, a prompt can be learned based on $\hat{x}_t$, thereby compensating for the bias $\Phi(\hat{x}_t, x_t)$ in Eq. (7) for the next model input. By adopting this approach, we can rectify the sampling trajectory by mitigating exposure bias at each time step, resulting in enhanced image quality.

### 4.1 Framework Training

As previously mentioned, based on the modeling Gaussian kernel $p_\theta(\hat{x}_t \mid x_{t+1})$, we aim to construct an anti-bias transition kernel $p(x_t \mid \hat{x}_t, x_{t+1})$ to match the true kernel $q(x_t \mid x_{t+1})$. To accomplish this goal, we employ a parameterized model $v_\phi(\cdot)$ to simulate $p(x_t \mid \hat{x}_t, x_{t+1})$. Below, we provide a detailed introduction on how to optimize $v_\phi(\cdot)$, with the conceptual framework shown in Figure 2.

**Overview** Based on the previous analysis, our prompt prediction model $v_\phi(\cdot)$ is designed to predict an anti-bias prompt that approximates the exposure bias defined in Eq. (7):

$$v_\phi(\hat{x}_t) \mapsto \Phi(\hat{x}_t, x_t), \tag{9}$$

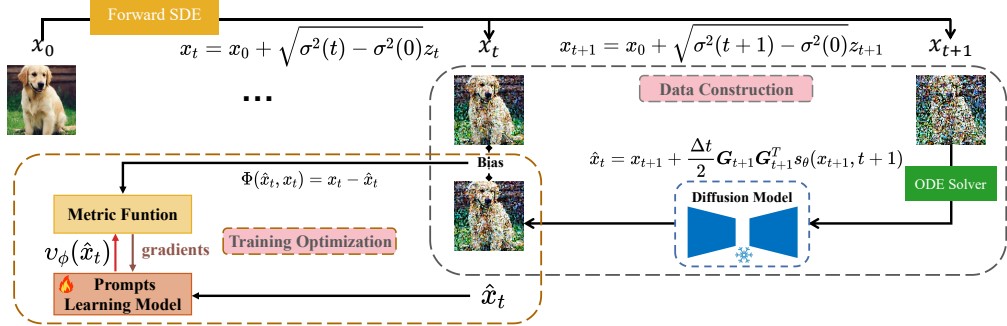

Figure 2: **Prompt Learning Framework Training.** For convenience, we employ the VE SDE and a first-order sampler to demonstrate the training process. 1) Data construction: obtain $x_t$ and $x_{t+1}$ via a forward SDE with randomly selected noises $\sigma_t$ and $\sigma_{t+1}$, and denoise $x_{t+1}$ to $\hat{x}_t$ with a deterministic sampler (*i.e.*, DDIM); 2) Training optimization: minimize the difference between the prompt $v_\phi(\hat{x}_t)$ and the exposure bias $\Phi(\hat{x}_t, x_t)$ via the training function Eq. (12), where $\Phi(\hat{x}_t, x_t)$ represents the discrepancy between the ground truth image $x_t$ and the modeling sample $\hat{x}_t$.

where $t$ is the time index of $x_t$. Since we do not adjust the parameters $\theta$ in the pre-trained DM, the term $p_\theta(\hat{x}_t \mid x_{t+1})$ remains fixed and can be used to generate $\hat{x}_t$. For a given prompt prediction model $v_\phi(\cdot)$, we have $v_\phi(\hat{x}_t) = \Phi(\hat{x}_t, x_t)$, where $t \in (0, T)$. Suppose we have a free-form deep neural network to represent our prompt prediction model, we can train it with our prediction loss:

$$\mathcal{L}(\phi, t) = \mathbb{E}\left[d(v_\phi(\hat{x}_t), \Phi(\hat{x}_t, x_t))\right] \tag{10}$$

and $d(\cdot, \cdot)$ is a metric function that satisfies $\forall x, y : d(x, y) \geq 0$ and $d(x, y) = 0$ if and only if $x = y$. The metric function is designed to minimize the difference between $v_\phi(\hat{x}_t)$ and $\Phi(\hat{x}_t, x_t)$. Inspired by this, there are several classic loss functions that satisfy our requirements, such as the squared $l_2$ norm $d(x, y) = \|x - y\|_2^2$ and the $l_1$ norm $d(x, y) = \|x - y\|_1$. Additionally, we also consider the contrastive loss (He et al., 2020; 2022) to maximize the similarity between the prompt and the bias. This loss has been successfully used in recent works on training or fine-tuning DMs, thanks to its theoretical guarantees (Daras et al., 2024; Zhang et al., 2024), with details provided in the Appendix.

**Bias Simulation** In practice, accessing $x_t$ is intractable because we cannot obtain the ground truth sampling trajectory. We only have the bias sample $\hat{x}_t$ due to the accumulation of exposure bias. Therefore, to address this issue, we seek help from the deterministic solver (Song et al., 2020a; 2021b) to simulate the bias $\Phi(\hat{x}_t, x_t)$. More concretely, we first utilize a forward SDE to perturb the target image $x_0$, allowing us to obtain the unbiased samples $x_{t+1}$ and $x_t$. For simplicity, we use the variance-exploding (VE) SDE (Song et al., 2021b) to illustrate this process:

$$x_{t+1} = x_0 + \sqrt{\sigma^2(t+1) - \sigma^2(0)}z_{t+1}, \tag{11}$$

where $\sigma(t+1)$ is the noise at time $t+1$, and $z_{t+1}$ is sampled from $\pi$, detailed in (Song et al., 2021b). Similarly, we can obtain $x_t$ by replacing $\sigma(t+1)$ with $\sigma(t)$ in Eq. (11). Subsequently, we employ the deterministic solver to denoise $x_{t+1}$ for just one time step, the detailed process is presented by Eq. (6). In this way, we have successfully simulated the exposure bias $\Phi(\hat{x}_t, x_t) = x_t - \hat{x}_t$ that arises at time step $t$. This is because the exposure bias problem is primarily caused by score estimation and discretization errors, both of which are simulated within one time step in Eq. (6). Although the deterministic solver may not fully denoise $\hat{x}_{t+1}$ to the final image, our goal is to model this deviation to rectify the sampling trajectory. Thus, we simulate exposure bias in a reasonable manner.

**Optimization** To optimize the prompt prediction model $v_\phi(\hat{x}_t)$, we utilize Eq. (10) to train the parameters $\phi$. However, in practice, the discrepancy between $v_\phi(\hat{x}_t)$ and $\Phi(\hat{x}_t, x_t)$ can be substantial across different time steps, resulting in irregular fluctuations in the training loss. This is due to the fact that noisy samples at different noise scales contain entirely different structural information. For instance, samples with low noise levels may provide preferable detailed information (Lou & Ermon, 2023), while samples with high noise levels may only capture coarser shapes.

Motivated by this observation, we propose a time-dependent weighting schedule designed to enhance training stability. To be specific, we utilize the signal-to-noise ratio (SNR) (Kingma et al.,

---

**Algorithm 1:** Anti-Bias Sampling

---

**Data:** pre-trained DM $s_\theta$, optimized prompt prediction model $v_{\phi^*}$, default sampler $S$, pre-defined noise schedule $L = \{\sigma_{t_0}, \ldots, \sigma_{t_T}\}$, total sampling steps $T$

**Result:** New Images $x_{t_0}^{\text{anti-bias}}$

1   sample a batch of $x_T$ from a prior distribution $\pi$;
2   $x_{\text{temp}} = x_T$;
3   **for** $t_i \leftarrow t_T$ **to** $t_0$ **do**
4      $\hat{x}_{t_i} = S(s_\theta, \sigma_{t_i}, x_{\text{temp}})$;
5      $x_{t_i}^{\text{anti-bias}} = v_{\phi^*}(\hat{x}_{t_i}) + \hat{x}_{t_i}$;   This anti-bias rectification is the only difference compared to the original sampling schedule.
6      $x_{\text{temp}} = x_{t_i}^{\text{anti-bias}}$;
7   $x_{t_0}^{\text{anti-bias}} = x_{\text{temp}}$;

---

2021; Choi et al., 2022) to formulate our weighting function, which is based on the coefficients of the forward SDE. The forward diffusion kernel can be represented as $q(x_t \mid x_0) = \mathcal{N}(\alpha_t x_0, \sigma_t^2 I)$, and therefore, our weighting function can be expressed as $\text{SNR}(t) = \alpha_t^2/\sigma_t^2$. In practice, for a given forward SDE, both $\alpha_t$ and $\sigma_t$ can be derived from the diffusion kernel $q(x_t \mid x_0)$. For example, in the variance preserving (VP) SDE (Song et al., 2021b; Ho et al., 2020), $\alpha_t = \sqrt{1 - \sigma_t^2}$, while in the variance exploding (VE) SDE (Song et al., 2021b; Song & Ermon, 2019), $\alpha_t = 1$. With the newly proposed weighting function, our training loss can be expressed as:

$$\mathcal{L}(\phi, \text{SNR}(t)) = \mathbb{E}\left[\text{SNR}(t)d(v_\phi(\hat{x}_t), \Phi(\hat{x}_t, x_t))\right]. \tag{12}$$

During the training process, we employ stochastic gradient descent on the model parameters $\phi$ via minimizing $\mathcal{L}(\phi, \text{SNR}(t))$, and updating $\phi^-$ with exponential moving average (EMA). We perform the following update with EMA after each training iteration: $\phi^- \longleftarrow \text{stopgrad}(\mu\phi^- + (1 - \mu)\phi)$. Here, $\mu$ is a decay rate with $0 \le \mu < 1$ (Song et al., 2023), with details provided in Appendix. When implementing these training techniques, we can effectively optimize the prompt prediction model.

## 4.2 ANTI-BIAS SAMPLING

Once the prompt prediction model $v_{\phi^*}(\cdot)$ is optimized, it can be used to improve sampling performance by reducing the exposure bias in the input of the pre-trained DMs for future steps, based on the bias predicted for the output of the model at the current step. To be specific, we utilize $v_{\phi^*}(\cdot)$ to predict an anti-bias prompt using the input $\hat{x}_t$, and the anti-bias image can thus be expressed as:

$$x_t^{\text{anti-bias}} = v_{\phi^*}(\hat{x}_t) + \hat{x}_t. \tag{13}$$

In the next time step, $x_t^{\text{anti-bias}}$ serves as the input of pre-trained DMs, allowing us to obtain $\hat{x}_{t-1}$ via Eq. (6). Subsequently, our model $v_{\phi^*}(\hat{x}_{t-1})$ predicts the prompt using the input $\hat{x}_{t-1}$. By iteratively implementing Eq. (6) and Eq. (13), $x_0^{\text{anti-bias}}$ can be generated with a high image quality, detailed shown in Figure 3 and Algorithm 1. Therefore, the prompting sampling trajectory can more closely match the training trajectory. Compared to the original diffusion sampling schedule, we retain the main procedure and only compensate a prompt for the output of a pre-trained DM at each time step.

On the other hand, our method can also improve the guidance sampling mechanism (Dhariwal & Nichol, 2021), which is a milestone technique to guide a sample with a pre-trained classifier $p(c \mid x_t, t)$, where $c$ represents a class label. The classifier guidance provides auxiliary information on the sampling trajectory by evaluating whether the sample is correctly classified according to the class label $c$. This is equivalent to sampling from the joint distribution $p(x_t, c)$ because:

$$\nabla \log p(x_t, c) = \nabla \log p(x_t) + \nabla \log p(c \mid x_t) \approx s_\theta(\hat{x}_t, t) + \nabla \log p(c \mid \hat{x}_t).$$

However, due to the presence of exposure bias, the biased image $\hat{x}_t$ may lead to inaccuracies in classification. This, in turn, results in a biased gradient $\nabla \log p(c \mid \hat{x}_t)$. Based on the previous analysis, our method enables further improvement in guided sampling via alleviating exposure bias:

$$\nabla \log p(x_t, c) = s_\theta(\hat{x}_t + v_{\phi^*}(\hat{x}_t), t) + \nabla \log p(c \mid \hat{x}_t + v_{\phi^*}(\hat{x}_t)). \tag{14}$$

It is worth noting that visual prompting method is indeed beneficial for image classification task (Jia et al., 2022; Bahng et al., 2022). Therefore, using a prompt prediction model to guide the score

Table 1: **Performance on CIFAR-10.** Here, we select NSCNv2 (Song & Ermon, 2020), DDPM (Ho et al., 2020), SDE (VE) (Song et al., 2021b), SDE (deep, VE) (Song et al., 2021b), ADM (Dhariwal & Nichol, 2021; Ning et al., 2023b) and ADM-IP (Ning et al., 2023b), as well as EDM (Karras et al., 2022) to serve as the baselines. When implementing our framework to them, the optimized prompt prediction models facilitate significant improvements in image quality, as evidenced by lower FID scores and better IS results. Notably, we use the original samplers proposed by the baselines, with the only difference being bias rectification, as shown in Algorithm 1.

| Models | FID↓ | IS↑ | NFEs↓ |
|---|---|---|---|
| NSCNv2 | 10.87 | 8.40 | 1000 |
| NSCNv2+ours | 9.56 | 8.65 | 1000 |
| DDPM | 3.17 | 9.46 | 1000 |
| DDPM+ours | 2.99 | 10.01 | 1000 |
| SDE (VE) | 2.55 | 9.83 | 1000 |
| SDE (VE)+ours | 2.41 | 9.91 | 1000 |
| SDE (deep, VE) | 2.20 | 9.89 | 1000 |
| SDE (deep, VE)+ours | 2.10 | 9.99 | 1000 |
| ADM | 3.56 | - | 100 |
| ADM+ours | 3.28 | - | 100 |
| ADM-IP | 3.12 | - | 100 |
| ADM-IP+ours | 3.06 | - | 100 |
| EDM | 2.04 | 9.84 | 35 |
| EDM+ours | 1.91 | 9.94 | 35 |

Table 2: **Connection to Training-free Fast Samplers.** Results obtained from experiments conducted on CIFAR-10. Here, we employ several classic samplers as our baselines, such as DDIM (Song et al., 2020a), Analytic-DPM (Bao et al., 2022b), DEIS (Zhang & Chen, 2022) and DPM-Solver (VP) (Lu et al., 2022b). For fairness, we use the same DMs as those used by the samplers in their papers. We confirm that our framework reduces the exposure bias caused by fast samplers with large step sizes, and the results are tested across various NFEs using FID↓.

| NFEs | 10 | 20 | 50 |
|---|---|---|---|
| DDIM | 13.36 | 6.84 | 4.67 |
| DDIM+ours | 12.94 | 6.71 | 4.59 |
| Analytic-DPM | 14.4 | 6.87 | 4.15 |
| Analytic-DPM+ours | 13.98 | 6.76 | 4.10 |
| DEIS (VP) | 4.17 | 2.86 | 2.57 |
| DEIS (VP)+ours | 4.08 | 2.80 | 2.51 |
| DEIS (VE) | 20.89 | 16.59 | 16.31 |
| DEIS (VE)+ours | 19.76 | 16.21 | 16.08 |

| NFEs | 12 | 24 | 48 |
|---|---|---|---|
| DPM-Solver-2 | 5.28 | 3.02 | 2.69 |
| DPM-Solver-2+ours | 5.22 | 2.95 | 2.65 |
| DPM-Solver-3 | 6.03 | 2.75 | 2.65 |
| DPM-Solver-3+ours | 5.93 | 2.69 | 2.61 |

direction is a reasonable approach, as it can significantly enhance the guidance gradient. Moreover, our framework has great potential for controllable generation (Ruiz et al., 2023; Nichol et al., 2022; Ramesh et al., 2022) via replacing $c$ to a text prompt, we leave this exploration for future work.

### 4.3 PROMPTING LATENT DIFFUSION

Recently, latent diffusion models (LDM) (Rombach et al., 2022; Peebles & Xie, 2023) have significantly enhanced the performance in image generation task. They employ an encoder $\mathcal{E}$ to map training images into latent representations $z = \mathcal{E}(x)$, and the decoder $\mathcal{D}$ to reconstruct the image from the latent $z$ with $\hat{x} = \mathcal{D}(\mathcal{E}(x))$. Given their promising future (He et al., 2023; Poole et al., 2022), it is meaningful to further enhance their generative performance using our framework.

Though their sampling trajectories traverse latent space, the exposure bias phenomenon still occurs due to the discrepancy between $z_t$ and $\hat{z}_t$. Here, $z_t$ represents the ground truth latent and $\hat{z}_t$ is the latent simulated by the pre-trained LDM. To remedy this, we put forward a variant of the prompt prediction model. Compared to diffusion in data space, the only difference is that we predict the prompt in the latent space. Concretely, the prompt sampling in latent space can be written as:

$$z_t^{\text{anti}-\text{bias}} = \upsilon_{\phi^*}(\hat{z}_t) + \hat{z}_t,$$

where $\upsilon_{\phi^*}(\cdot)$ can be optimized via minimizing the metric function $d(\upsilon_\phi(\hat{z}_t), \Phi(\hat{z}_t, z_t))$ using a gradient descent algorithm. In this manner, our prompt learning framework effectively reduces the bias between $\hat{z}_t$ and $z_t$, thereby contributing to improved sampling trajectories in latent diffusion.

### 5 EXPERIMENTS

To evaluate the effectiveness of our prompt learning framework in reducing exposure bias, we conduct experiments on three benchmark datasets: CIFAR-10 (Krizhevsky et al., 2009), CelebA

Table 3: **Performance on ImageNet** $256 \times 256$. We select ADM (Dhariwal & Nichol, 2021) and ADM-U (Dhariwal & Nichol, 2021) as baselines. After applying the optimized prompt model to the default sampler in ADM, both achieve improvements in image quality.

| Models | FID↓ | IS↑ | NFEs↓ |
|---|---|---|---|
| ADM | 10.94 | 100.98 | 250 |
| ADM+ours | 10.37 | 112.00 | 250 |
| ADM-U | 7.49 | 127.49 | 250 |
| ADM-U+ours | 7.29 | 134.95 | 250 |

Table 4: **Boosting Latent Diffusion.** We use LDM (Rombach et al., 2022) as our baseline and test its performance on ImageNet at a resolution of $256 \times 256$. Both LDM-4 and LDM-8 show significant improvements after applying the optimized model to rectify bias in the default LDM sampler.

| Models | FID↓ | IS↑ | NFEs↓ |
|---|---|---|---|
| LDM-4 | 10.56 | 103.49 | 250 |
| LDM-4+ours | 10.02 | 111.69 | 250 |
| LDM-8 | 15.51 | 79.03 | 200 |
| LDM-8+ours | 14.03 | 91.02 | 200 |

$64 \times 64$ (Liu et al., 2015), and ImageNet $256 \times 256$, utilizing various pre-trained DMs. Concretely, for CIFAR-10, we select NSCNv2 (Song & Ermon, 2020), DDPM (Ho et al., 2020), SDE (VE) and SDE (deep, VE) (Song et al., 2021b), ADM (Ning et al., 2023b) and ADM-IP (Ning et al., 2023b), as well as EDM (Karras et al., 2022) to serve as the baseline models. For CelebA, we utilize ADM (Dhariwal & Nichol, 2021; Ning et al., 2023b) and ADM-IP (Ning et al., 2023b) as baseline models. In contrast, for ImageNet, we select ADM (Dhariwal & Nichol, 2021) and ADM-U (Dhariwal & Nichol, 2021) to serve as the baseline models. We then employ prompt models customized for different DMs to mitigate exposure bias at each step, aiming to enhance image quality. To quantitatively evaluate the performance of our framework, we utilize standard metrics, including Fréchet Inception Distance (FID) (Heusel et al., 2017), Inception Score (IS) (Salimans et al., 2016) and Spatial Fréchet Inception Distance (sFID) (Nash et al., 2021), as well as neural function evaluations (NFEs) (Vahdat et al., 2021), to verify them on 50K newly generated samples.

As mentioned previously, our framework demonstrates good theoretical flexibility, such as enhancing guidance mechanism, alleviating exposure bias in latent space, and improving image quality for training-free fast samplers. To verify this, we choose ADM-G (Dhariwal & Nichol, 2021) to assess the effectiveness of our framework on classifier guidance methods. Moreover, we also evaluate the performance of improving the latent diffusion model, thus choosing LDM-4 and LDM-8 (Rombach et al., 2022) as the baseline models. For evaluating the effectiveness on rectifying high-order solvers, we employ our framework to improve several classic training-free fast samplers.

To design the architecture of prompt prediction model, we employ a lightweight U-shaped network, with a backbone similar to that of EDM (Karras et al., 2022). We design the model architecture according to different data resolutions. Specifically, we set the model channels for resolutions 32, 64, and 256 as 32, 32, and 64, respectively. The corresponding model parameters are 3.2M, 3.2M, and 12.7M. We maintain these settings on all experiments, more training details refer to the Appendix.

## 5.1 PERFORMANCE EVALUATION

**Quantitative Comparison** To evaluate performance in the data space, we conduct multi-group experiments on various datasets. In Table 1, we first present the evaluation of two well-known DMs in discrete diffusion, as well as the classic SDE DM, all of which demonstrate significant improvements. Importantly, we also demonstrate further improvements on the pioneering work of ADM-IP (Ning et al., 2023b), the ADM-IP, which first investigates the exposure bias problem. For performance evaluation on CelebA and ImageNet, we utilize ADM (Dhariwal & Nichol, 2021) and ADM-IP (Ning et al., 2023b) for verification, with results depicted in Table 3 and Table 8, respectively. On the other hand, our framework also possesses the capacity to reduce exposure bias in latent diffusion, demonstrating great flexibility. To be specific, our customized models for latent diffusion enable to improve LDM-4 and LDM-8 in terms of FID and IS, as detailed in Table 3. Moreover, we also test the effectiveness of our framework in enhancing guidance sampling, as shown in Table 4. This is because lower bias samples enable better classification accuracy, thus providing more efficient classifier gradients. In this context, our framework naturally improves generative performance when combined with guidance methods. As mentioned earlier, the exposure bias is notably larger in cases with fewer NFEs because the discretization error increases with a larger sampling step size.

Table 5: **Enhancing Guidance Mechanism.** We evaluate performance in a guidance setting using unconditional ADM-G (Dhariwal & Nichol, 2021) with ImageNet at a resolution of 256 × 256. Here, "Scale" indicates the degree of guidance provided by the classifier. After applying our prompt prediction model, they enable to achieve improvements in image quality, as reflected by better FID and higher IS scores. All results are based on 50k samples generated using 250 NFEs.

| Models | Scale | FID↓ | IS↑ |
|---|---|---|---|
| ADM-G | 1.0 | 33.03 | 32.92 |
| ADM-G+ours | 1.0 | 31.64 | 36.00 |
| ADM-G | 10.0 | 12.00 | 95.41 |
| ADM-G+ours | 10.0 | 10.64 | 106.13 |

Table 6: **Ablation Studies on Different Metric Functions.** Here, we employ EDM (Karras et al., 2022) to test three different metric functions, including contrastive loss, $L_1$ norm, and $L_2$ norm, to train our prompt prediction model on CIFAR-10. For fairness, we set the batch size to 1024 and the model channels to 32 in all experiments. Obviously, the contrastive loss presents enhanced performance due to its theoretical guarantees in the training bound.

| Metric Functions | FID↓ | NFEs↓ | Iterations |
|---|---|---|---|
| EDM (baseline) | 2.04 | 35 | - |
| $L_1$ norm | 1.96 | 35 | 100k |
| $L_2$ norm | 1.94 | 35 | 100k |
| Contrastive loss | 1.96 | 35 | 80k |
| Contrastive loss | 1.93 | 35 | 100k |
| Contrastive loss | 1.91 | 35 | 150k |

When implementing our framework with fast samplers such as DDIM (Song et al., 2020a), Analytic-DPM (Bao et al., 2022b) and DEIS (VP) (Zhang & Chen, 2022), as well as DPM-Solver (Lu et al., 2022b), the resulting sampling trajectory yields good results, as detailed in Table 2. Moreover, Table 10 presents the efficiency of our model in comparison with the settings by using more NFEs.

**Qualitative Comparison** To demonstrate the effectiveness of our approach from a qualitative perspective, we present some visualization results in Figure 7. In Figure 7, the generated images exhibit rich semantic information along with vivid visual effects. For a thorough validation of performance in alleviating exposure bias, we present side-by-side visual comparisons in the Appendix.

**Ablation Study** To evaluate the effectiveness with different metric functions, we conduct ablations on CIFAR-10 with EDM. For fairness, we train the prompt prediction model under the same settings. In this context, our model achieves great improvements on EDM, as shown in Table 6. In particular, the contrastive loss even slashes the FID from 2.04 to 1.91, effectively increasing the anti-bias performance regarding image quality. Though $L_1$ norm and $L_2$ norm do not achieve such remarkable improvements on EDM, they effectively validate the performance of our framework. Thus, our framework is validated in its ability to improve image quality by alleviating exposure bias.

## 6  CONCLUSION

In this paper, we conduct an in-depth investigation into the training-sampling discrepancy, referred to as exposure bias, which arises from score estimation and discretization errors. To alleviate exposure bias and thereby improve image quality, we put forward a prompt learning framework that employs a lightweight parameterized model to compensate for the bias. The optimized prompt prediction model can improve various pre-trained DMs on different beachmark datasets, with the additional sampling overhead being less than 5%. Moreover, our framework demonstrates great flexibility in adapting to various DM settings, including guidance mechanism, latent diffusion, and fast samplers.

**Broader Impacts and Limitations** While our method has achieved significant improvements, it still requires overhead for training and sampling. Besides, it is important to acknowledge that generating deepfake images using our model also entails the potential risk of negative misuse of this technology.

## ACKNOWLEDGMENTS

This research was supported by the Ministry of Science and ICT (MSIT) of Korea, under the National Research Foundation (NRF) grant (RS-2024-00337548). This work was supported in part by the Australian Research Council under Projects DP240101848 and FT230100549. This work was supported by the Natural Science Foundation of China under grant 61836016. The AI training platform supporting this work were provided by High-Flyer AI.

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

## A   RELATED WORKS

**Diffusion Models** DMs (Dhariwal & Nichol, 2021; Ho et al., 2020; Kingma et al., 2021; Nichol & Dhariwal, 2021) are a new family of generative models with remarkable performance, particularly in the field of 2D image generation (Gao et al., 2023; Vahdat et al., 2021; Rombach et al., 2022), due to their ability to model complex data distributions. This is mainly because DM training directly models the target data distribution via minimizing the upper bound of the model log likelihood (Sohl-Dickstein et al., 2015; Luo, 2022). In this manner, DMs enable to achieve comparable mode coverage (Kingma et al., 2021; Song et al., 2021a; Lu et al., 2022a; Kim et al., 2022), reflected at lower negative log likelihood. Based on the rigorous SDE framework (Song et al., 2021b; Anderson, 1982), some classic DMs contribute the image quality from different aspects, including deeper model structure (Song & Ermon, 2020; Song et al., 2021b; Kingma et al., 2021; Peebles & Xie, 2023; Kim et al., 2023b), diffusion schedule (Lin et al., 2024), diffusion in latent space (Vahdat et al., 2021; Rombach et al., 2022; Jing et al., 2022), refined weighting schedules (Choi et al., 2022; Kim et al., 2022; Song et al., 2021a) and well-designed training objectives (Kingma et al., 2021; Nichol & Dhariwal, 2021; Karras et al., 2022), as well as rational optimization strategies (Hang et al., 2023; Wu et al., 2023). Moreover, recent works utilize conditional information to guide the image generation (Dhariwal & Nichol, 2021; Ho & Salimans, 2022), such as class label and text prompt (Ramesh et al., 2022; Ruiz et al., 2023; Saharia et al., 2022; Li et al., 2023c), which further improve the image quality. With the help of those techniques, DMs achieve new SoTA modeling ability (Kim et al., 2023a; Peebles & Xie, 2023; Kim et al., 2023b) and better class diversity compared to previous SoTA generative models. Although they enable the generation of high-quality images, they cannot avoid the exposure bias problem, which does influence the image quality.

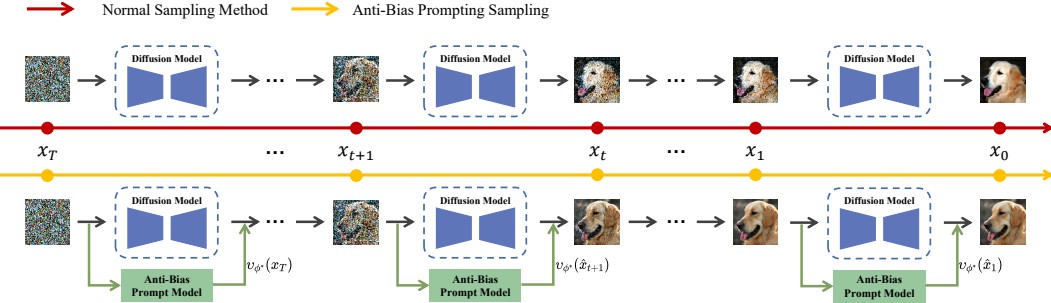

Figure 3: **Anti-Bias Sampling.** Our framework employs the prompt prediction model $v_{\phi^*}(\cdot)$ to predict a prompt with the input of $\hat{x}_{t+1}$, and compensate the bias in the current time step $t+1$. Better samples can be obtained via $v_{\phi^*}(\hat{x}_{t+1}) + \hat{x}_{t+1}$, serving the input of DMs in next time step $t$. We can therefore enhance the image quality by iteratively running this process.

**Exposure Bias Problem** The exposure bias problem is originally mainly studied for language models by the nature language process community (Ranzato et al., 2015), and the exposure bias of diffusion model is less explored (Ning et al., 2023b; Li et al., 2023a). Some classic works propose various methods to reduce the score estimation and discretization errors, i.e., the underlying source of the exposure bias, to handle the exposure bias problem indirectly. For instance, some methods make great efforts using high order numerical solvers (Zhang & Chen, 2022; Li et al., 2023b; Zhao et al., 2023; Zhang et al., 2023). Stable diffusion (Rombach et al., 2022) matches the score in latent space (Vahdat et al., 2021), which naturally reduce the discretization error by solving the integral in a lower dimension. DMCMC (Kim & Ye, 2022) utilizes MCMC to obtain a good initialization points close to the modeling distribution, aiming to reduce the accumulation of errors. Besides, (Kim et al., 2023a; Chao et al., 2022) propose to adjust the matched score via a robust discriminator. Beyond the usual treatment, certain works (Salimans & Ho, 2022; Song et al., 2023; Meng et al., 2023; Kim et al., 2023b) mitigate the sampling errors in small time steps of model sampling by distilling knowledge from larger sampling steps. Specifically, Consistency Model (Song et al., 2023) even slashes the neural function evaluations (NFEs) to only two steps with improved score matching accuracy. Recently, rectified flow (Liu et al., 2022; 2023c; Esser et al., 2024; Ma et al., 2024) simulates the optimal transport between prior distribution and target data distribution via straightening the sampling trajectory. Orthogonal to them, we present a prompt learning framework, which employs a transition function to learn an anti-bias prompt to compensate the next model input and handle the exposure bias in a direct manner.

**Prompt Learning** Prompt learning is first proposed in natural language processing (NLP) (Liu et al., 2023b), which employs a text to help pre-trained large models (LMs) "understand" the task. Subsequently, GPT-3 (Brown et al., 2020) demonstrates remarkable performance to downstream transfer learning tasks even in the shot or zero-shot settings (Radford et al., 2021; Ouyang et al., 2022). To improve the readability of prompting text for LMs, some methods (Jiang et al., 2020; Shin et al., 2020) propose constructing more plentiful prompting texts. Recently, some heuristic approaches consider a more efficient way, treating prompt as task-specific continuous vectors and fine-tuning them via gradient propagation, namely Prompt Tuning (Lester et al., 2021; Li & Liang, 2021; Liu et al., 2021). Based on the great success on LMs, there are lots of vision LMs (He et al., 2021; Radford et al., 2021; Yao et al., 2021; Zhou et al., 2022) employ the text encoder to extract more information from the text prompt. More recently, Jia et al. (Jia et al., 2022) explored visual prompting in recognition tasks. As a concurrent work, Bahng et al. (Bahng et al., 2022) demonstrated that visual prompting is effective for CLIP and distribution shift. In this paper, we propose our anti-bias prompt learning, a novel variant of the visual prompting, aimed at alleviating the exposure bias problem in DMs.

## B EXPERIMENTAL DETAILS

**Architecture** We follow the EDM framework (Karras et al., 2022), which adopts the NCSN++ model proposed by (Song et al., 2021b) as the backbone of our prompt model, as shown in Table

8 of EDM paper. To be specific, NCSN++ is a U-shaped architecture based on (Ho et al., 2020) that uses Finite Impulse Response (FIR) upsampling and downsampling, rescales skip connections, and employs four BigGAN (Brock, 2018) residual blocks at each resolution. Moreover, NCSN++ incorporates additional residual skip connections from the input image to each block in the encoder. The only difference between NCSN++ and our prompt model backbone is that we remove the time embedding and reduce the number of model channels for greater efficiency. Concretely, we set the model channels in the NCSN++ backbone for resolutions 32, 64, and 256 to 32, 32, and 64, respectively. As a result, the corresponding model parameters for the different prompt models are 3.2M, 3.2M, and 12.7M.

**Training** As mentioned before, we employ the EDM backbone to serve as our prompt prediction model and train our prompt model on the same datasets used to train the baseline models for a fair comparison. In practice, we exclude the time embedding setting from the architecture of EDM, thus reducing some model parameters. For instance, the original EDM backbone contains 56M parameters, whereas the backbone without time embedding contains 51M parameters. During training, we set the batch size to 1024 for all experiments and keep other hyperparameters the same as in EDM training. Detailed training settings can be found in (Karras et al., 2022), and we maintain the default values. To train the model, we allocate A100 GPUs to optimize them and test the experimental results on just one A100 GPU. Specifically, we employ 8 A100 GPUs for training CIFAR-10 and CelebA, while we use 16 A100 GPUs for training on ImageNet. Additionally, we allocate only 4 A100 GPUs for training the prompt prediction model for latent diffusion. For most experiments, the training iterations range from 100k to 150k across all datasets, which are considered acceptable training expenses. In practice, for the EMA selection, we maintain the same settings as those used in the consistency models (Song et al., 2023). Specifically, we set the EMA value to 0.9999 when training our model on CIFAR-10, and we set the EMA to 0.999943 for LSUN and ImageNet.

**Sampling** After completing the training process, we employ the optimized model to reduce exposure bias using the sampling process shown in Figure 3 and Algorithm 1. It is worth noting that if the total sampling NFEs for the original sampler is $T$, our prompt prediction model will be employed $T-1$ times within the same sampler when aiming to mitigate exposure bias. For sampling computation, our prompt prediction model only increase the sampling time less than 5%. To verify this, we test it on an A100 GPU via sampling 1k images with 35 NFEs. Concretely, EDM requires 33.9 seconds, while our model increases the time to only 36.3 seconds, with a cost increase of 2.4 seconds. When sampling one image, the overhead can be almost ignored. For side-by-side comparison, we present more results in Figure 4 to Figure 6. Other results are shown in Figure 7 and Figure 8, all are randomly generated. Moreover, we present an additional ablation study to evaluate performance with different model parameters, as shown in Table 9. In this paper, we employ the backbone with 3.2M parameters to serve as our prompt model, to ensure the efficiency in sampling process.

**Comparison on larger model or more sampling steps** We also conduct experiments to test whether using more NFEs or additional parameters can achieve better results, detailed shown in Table 10. While employing a larger model has the potential to decrease score estimation errors, it necessitates substantial training resources and considerable human effort. This is because the exposure bias phenomenon is caused by inherent factors within the diffusion modeling framework. Analogously, increasing NFEs will improve the image quality, but the performance marginally improves and does not increase indefinitely. Hence, our framework is meaningful for the diffusion community as it provides a special case for reducing exposure bias. Furthermore, we test more metrics to verify the effectiveness of our model, including sFID and NFEs, detailed shown in Table 7. Specifically, we achieve improvements on both CIFAR-10 and ImageNet when using our model to enhance ADM-IP.

**Diversity Testing** Following the main design philosophy, our model can improve the image quality without affecting the image diversity. Because the diversity in diffusion modeling framework is mainly depends on the diffusion term $\boldsymbol{G}_t z_t$ in Eq. (1). On the contrary, our model only change the score term to $s_\theta(\hat{x}_{t+1} + \upsilon_{\phi^*}(\hat{x}_{t+1}), t+1)$, thus our model will not affect diversity. To verify this, we also conduct experiments to test precision and recall on ADM, which are common metrics for evaluating diversity. Specifically, the precision and recall values are 0.69 and 0.63, respectively. These values are still 0.69 and 0.63 after employing our framework. Since precision and recall remain the same, our model has no negative effects on the diversity.

**Comparison with training free anti-bias model** We conducted experiments on the recent training-free method (Ning et al., 2023a), which is effective in reducing the exposure bias without any train-

Table 7: **More Metrics for Evaluating Performance in Comparison with ADM-IP.** To better test the effectiveness of our prompt prediction model, we employ more metrics, such as sFID↓ and NFEs↓, to test it on different datasets, including CIFAR-10 and ImageNet. Moreover, we employ the classic ADM-IP (Ning et al., 2023b), the first work to address the exposure bias issue, to verify these results. As shown below, our prompt model further reduces the exposure bias for ADM-IP on these datasets with different NFEs, despite ADM-IP being designed with a training strategy to mitigate exposure bias, demonstrating the flexibility of our model.

| Models | Dataset | sFID↓ | NFEs↓ |
|---|---|---|---|
| ADM-IP | CIFAR-10 | 3.86 | 100 |
| ADM-IP+Ours | CIFAR-10 | 3.80 | 100 |
| ADM-IP | CIFAR-10 | 3.89 | 80 |
| ADM-IP+Ours | CIFAR-10 | 3.84 | 80 |
| ADM-IP | ImageNet | 3.11 | 100 |
| ADM-IP+Ours | ImageNet | 3.04 | 100 |
| ADM-IP | ImageNet | 3.36 | 80 |
| ADM-IP+Ours | ImageNet | 3.33 | 80 |

Table 8: **Performance on CelebA.** To evaluate effectiveness, we employ ADM (Dhariwal & Nichol, 2021) and ADM-IP (Ning et al., 2023b) as baseline models, using the default samplers from the original papers.

| Models | FID↓ | NFEs↓ | sFID↓ |
|---|---|---|---|
| ADM | 3.02 | 100 | 5.76 |
| ADM+ours | 2.93 | 100 | 4.74 |
| ADM-IP | 2.21 | 100 | 4.33 |
| ADM-IP+ours | 2.15 | 100 | 4.19 |

Table 9: **Ablation Study on Model Parameters.** We design various prompt models (PM) with different parameters to enhance EDM (Karras et al., 2022) on CIFAR-10, the detailed FID↓ results are shown as below.

| Models | Parameters | FID↓ |
|---|---|---|
| EDM | - | 2.04 |
| EDM+PM (large) | 12.7M | 1.90 |
| EDM+PM (regular) | 3.2M | 1.91 |
| EDM+PM (small) | 0.8M | 2.02 |

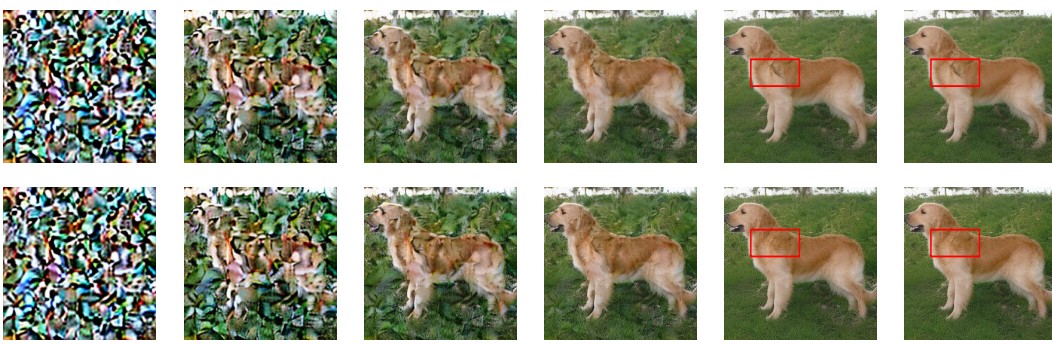

Figure 4: **Side-by-Side Comparison.** We use ADM trained on ImageNet 256 as the baseline, with the sampler set to DDIM. The first row displays images generated by ADM, while the second row shows images improved by our prompt model, with both rows using the same initial noise. The difference is highlighted by a red box, indicating that our model can mitigate unrealistic features.

ing. The FID values for EDM and EDM-ES presented in (Ning et al., 2023a) are 1.97 and 1.95, respectively, because they were calculated using a fixed seed. For a fair comparison, we recalculated the FID using a random seed, resulting in values of 2.04 and 2.01. Compared to EDM-ES, our prompt model significantly improves generative performance, as the FID decreases from 2.04 to 1.91. Although our method achieves much better performance than EDM-ES, it requires additional training computation, which is unfair for comparison. Hence, we do not present it in our main paper.

## C PROOFS OF CONTRASTIVE LOSS

In our experiments, the best results are achieved by using contrastive loss as the metric function. This is mainly because the contrastive loss theoretically assured that can further reduce the KL divergence $D_{\mathrm{KL}}\left(p(x_t \mid x_{t+1}) \| p_\phi(x_t \mid \hat{x}_t, x_{t+1})\right)$ between the true Gaussian kernel $p(x_t \mid x_{t+1})$ and the modeling kernel $p_\phi(x_t \mid \hat{x}_t, x_{t+1})$. Below, we provide a detail proof for this bound.

Table 10: Performance of EDM (Karras et al., 2022) with More NFEs and Model Parameters on CIFAR-10. To verify efficiency, we use more NFEs to test FID by sampling 50K images and report the corresponding time cost below.

| Models × NFEs | FID↓ | Time (min.)↓ |
|---|---|---|
| EDM × 35 | 2.04 | 28.0 |
| EDM × 35 + Ours × 34 | 1.91 | 30.3 |
| EDM × 37 | 2.03 | 30.1 |
| EDM × 47 | 2.02 | 38.3 |
| EDM × 57 | 2.02 | 45.7 |

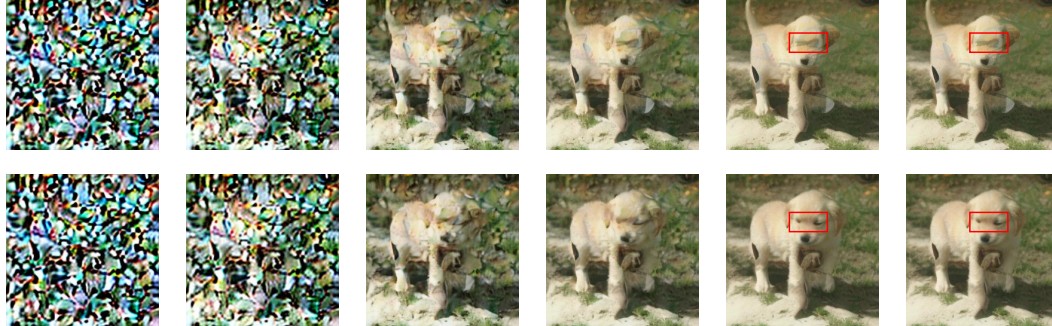

Figure 5: **Side-by-Side Comparison.** We use ADM trained on ImageNet 256 as the baseline, with the sampler set to DDIM. The first row displays images generated by ADM, while the second row shows images improved by our prompt model, with both rows using the same initial noise. The difference is highlighted by a red box, indicating that our model can mitigate unrealistic features.

**Theorem 1.** *Let $p(x_t)$ be the marginal probability density at time step $t$ that locates in the diffusion path, $p_\theta(\hat{x}_t)$ be the distribution from the reverse path that simulated by a pre-trained DM. Assume $X_t$ and $\hat{X}_t$ represent two batches images of $x_t$ and $\hat{x}_t$ that sampled from $p(x_t)$ and $p_\theta(\hat{x}_t)$ respectively. Then we can derive the upper bound of the gap between the true and the modeling transition kernels is actually the InfoNCE loss $\mathcal{L}_{\mathrm{InfoNCE}}(\cdot,\cdot)$*

$$D_{\mathrm{KL}}\left(p(x_t \mid x_{t+1}) \| p_\phi(x_t \mid \hat{x}_t, x_{t+1})\right) \leq \mathcal{L}_{\mathrm{InfoNCE}}(X_t, \hat{X}_t). \tag{15}$$

*Proof.* Before derive the Eq. (15), we first consider the mutual information (MI) Poole et al. (2019) between two batches of images $X_t$ and $\hat{X}_t$ at time step $t$, denoted as $I(X_t; \hat{X}_t)$. We can build a tractable variational upper bound by introducing the true distribution $p(x_t)$ in the diffusion path to the intractable marginal $p(x_t \mid x_{t+1}) = \int d\hat{x}_t p_\theta(\hat{x}_t \mid x_{t+1}) p_\phi(x_t \mid \hat{x}_t, x_{t+1})$. In theory, it is tractable to map $x_{t+1}$ to $x_t$ with the state transition matrix. By multiplying and dividing the integrand in MI by $p(x_t)$ and dropping a negative KL term, we enable to get the tractable variational upper bound (Poole et al., 2019): $I(X_t; \hat{X}_t) \geq D_{\mathrm{KL}}\left(p(x_t \mid x_{t+1}) \| p_\phi(x_t \mid \hat{x}_t, x_{t+1})\right)$. Analogously, by optimizing $\mathcal{L}_{\mathrm{InfoNCE}}(\cdot,\cdot)$, we can connect the InfoNCE loss with MI (Oord et al., 2018) via $\mathcal{L}_{\mathrm{InfoNCE}} \geq \log(N) - I(X_t; \hat{X}_t)$. Here, $N$ is the number of images in each training batch that containing one positive sample and $N-1$ negative samples. Remarkably, a large $N$ will make this bound tighter. In this context, we enable to derive the upper bound of $D_{\mathrm{KL}}\left(p(x_t \mid x_{t+1}) \| p_\phi(x_t \mid \hat{x}_t, x_{t+1})\right)$ is actually a well-designed contrastive loss $\mathcal{L}_{\mathrm{InfoNCE}}(\cdot,\cdot)$ via employing Jensen's inequality.

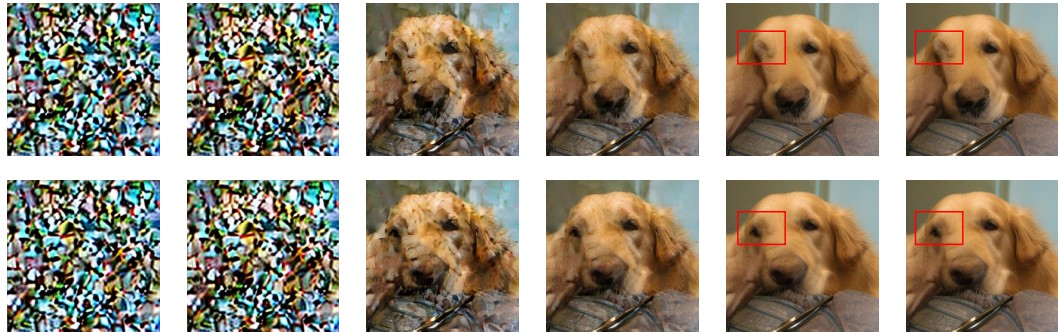

Figure 6: **Side-by-Side Comparison.** We use ADM trained on ImageNet 256 as the baseline, with the sampler set to DDIM. The first row displays images generated by ADM, while the second row shows images improved by our prompt model, with both rows using the same initial noise. The difference is highlighted by a red box, indicating that our model can mitigate unrealistic features.

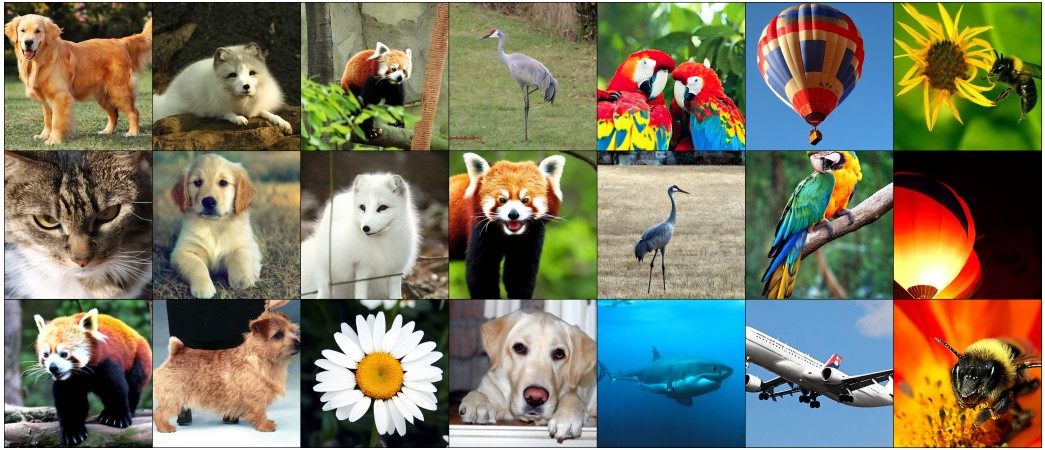

Figure 7: Randomly selected 256×256 images improved by our prompt learning framework.

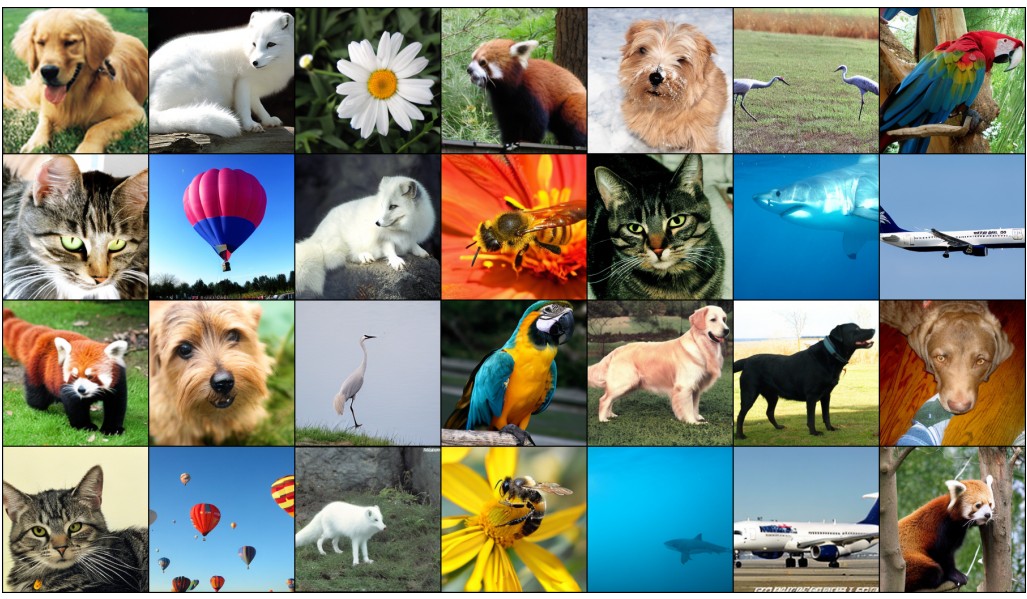

Figure 8: Randomly selected 256×256 images improved by our prompt learning framework.

