# OpenReview forum: "Anti-Exposure Bias in Diffusion Models"
_ICLR.cc/2025/Conference — ICLR 2025 Spotlight_

### Official Review · Reviewer_2SrK · 2024-10-18

**Soundness:** 3
**Presentation:** 3
**Contribution:** 3
**Rating:** 6
**Confidence:** 4

**Summary:**

This paper analyzes the gap between the true score function during training and the score function estimated during inference, highlighting that this gap can be narrowed, thereby improving model performance through the training of a prompt learning model. Experiments demonstrate that the algorithm performs well on both 64x64 and 256x256 diffusion models.

**Strengths:**

1. This work revisits the gap in the score function during training and inference, and proposes a fine-tuning framework to mitigate this gap.

2. The idea of this work is novel, which suggests that it is difficult to achieve perfect estimation of the score function when diffusion models are trained.

3. The presentation of the paper was excellent and easy to understand.

4. The authors conduct sufficient experiments across various diffusion models, including those on CIFAR-10, ADM on ImageNet-256x256, and Latent Diffusion, to support the effectiveness of the algorithm.

**Weaknesses:**

1. while the authors use a lot of analysis to show that the gap between the true Gaussian kernel and its modeling counterpart does exist, there is a lack of visualization to further illustrate this point.

2. It is suggested to move Table 9 in the appendix to the main paper, this is important to illustrate the efficiency of the algorithm.

3. The authors need to conduct further analysis of the effect of the number of parameters of the Anti-Bias Prompt Model on inference performance.

4. The authors do not provide code, which is important for the reproduction of the algorithm, even if it is simple code on CIFAR-10.

**Questions:**

The authors utilize the learning paradigm of the Prompt Learning Model and inquire whether they have attempted to minimize the difference in Eq. 12 using LoRA. If so, there may not be any additional computational overhead required?

---

### Official Review · Reviewer_3M1k · 2024-10-27

**Soundness:** 2
**Presentation:** 2
**Contribution:** 2
**Rating:** 8
**Confidence:** 3

**Summary:**

This paper proposes to learn an anti-bias prompt for the generated sample at each sampling step, aiming to compensate for the exposure bias that arises. Following this design philosophy, our framework rectifies the sampling trajectory to match the training
trajectory, thereby reducing the divergence between the target data distribution and the modeling distribution.

**Strengths:**

1. The proposed method aims to solve a challenging problem known as exposure bias, which has a wide range of application scenarios.
2. The proposed method is theoretically sound.

**Weaknesses:**

1. More visualization comparision should be provided.
2. Based on the visual results, the improvement of the proposed method is not very significant.

**Questions:**

See the weakness.

---

### Official Review · Reviewer_Pxp6 · 2024-10-27

**Soundness:** 3
**Presentation:** 4
**Contribution:** 3
**Rating:** 8
**Confidence:** 3

**Summary:**

This paper addresses the exposure bias in diffusion models caused by score estimation errors and discretization errors, by proposing a novel prompt learning framework. This framework employs a prompt prediction model that learns anti-bias information in each timestep during the reverse process, thereby mitigating the impact of exposure bias during image generation. To train this prompt prediction model, the authors utilize the Variational Estimation Stochastic Differential Equation (VE SDE) to simulate exposure bias. Extensive experiments were conducted on several traditional benchmarks, including CIFAR, CelebA, and ImageNet 256x256, demonstrating significant improvements across various diffusion models.

**Strengths:**

1. This paper is well-structured and presents its findings effectively. The authors provide substantial reasoning for each step, making it an enjoyable read that offers valuable insights.

2. The paper introduces a novel prompt learning framework that addresses the gap between training and sampling in diffusion models by training an additional model. The $\textbf{prompt prediction model}$ effectively mitigates exposure bias, and the authors conducted extensive experiments on several well-known benchmark datasets, yielding significant results. Furthermore, the model introduces minimal time delay, demonstrating practical value.

**Weaknesses:**

Overall, from my perspective, I think this paper is valuable and should be accepted. However, there still remains some weaknesses. If the authors can respond my puzzles, I will raise my scores.

For presentation:

1. The title of this paper ``$\textbf{Anti-Exposure Bias in Diffusion Models via Prompt Learning}$''. Through the whole paper, I think it is more like noise learning instead of prompt learning? What is the relationship between the proposed methods and prompt learning?

2. This paper only provides the improved images in Fig.7. Can the authors provide more images for initutive comparsions?

For method:

1. In eqn. 7, the authors claim $\Phi(\hat{x}_t,x_t)$ represents the exposure bias. However, based on my understanding, I only find the $\textbf{discretization error}$, rather than both $\textbf{discretization error}$ and $\textbf{score estimation error}$.

2. This paper utilizes VE SDE to get $x_{t+1}$, in order to simulate exposure bias. However, from my perspective, I think the $\hat{x}_t$ from prompt prediction model is not the Gaussian noise. This approach can be considered as a way to mitigate the training-sampling discrepancy of diffusion model such as [1]. Any comparsions between these methods?

[1] Common Diffusion Noise Schedules and Sample Steps are Flawed

For experiments:

1. All of these experiments are conducted in traditional benchmark, such as CIFAR, CelebA, and ImageNet 256x256. Moreover, the DMs used in this paper are not the popular T2I diffusion models, such as SDXL. Although the authors leave this as the future work, I think without these experiments, the practial value of this paper is limited.

2. Moreover, in this paper, the authors state the batch size of all experiments is $\textbf{1024}$, and the iteration of EDM in Tab. 8 requires 150k. That means approximate 100M data samples. The  $\textbf{prompt prediction model}$ is lightweight and the parameters of the DMs are fixed. Why such a huge amount of data samples are needed?

3. Based on 2, the size of image in all experiments are not bigger than 256x256. How about generating the images with 1024x1024, should it require more data samples and computional resources?

**Questions:**

See the weaknesses.

---

> ### Comment · Reviewer_Pxp6 · 2024-11-23
>
> Thanks to the author's response.
>
> For P2. More imgaes for initutive comparsions.
>
> The Figure 7 and Figure 8 use the same caption. I have no idea about which one is the original result.
>
> Most of my concerns were addressed. Good luck!

---

### Official Review · Reviewer_TaPC · 2024-11-01

**Soundness:** 3
**Presentation:** 3
**Contribution:** 3
**Rating:** 6
**Confidence:** 4

**Summary:**

In this paper, the authors begin by addressing the exposure bias that becomes increasingly pronounced along the sampling trajectory, as each time step accumulates newly generated score estimation and discretization errors. To tackle this issue, the authors propose a prompt learning module designed to eliminate the bias introduced by noised images from previous steps during the training and inference processes. The experiments demonstrate the effectiveness of the prompt learning across various model architectures and datasets.

**Strengths:**

1.The insight is interesting.

2.The method is simple and effective.

3.The author presents some theoretical analysis which makes the paper more believable.

**Weaknesses:**

1.The architecture of the prompt learning network is unclear. It is recommended that the authors include the proposed module architecture or network detail description in the supplementary materials to aid readers in understanding the method.

2.While the experiments indicate some improvement from the proposed module, the selection of baseline models is relatively simple. Will the issue of exposure bias become less significant in practical applications after updating the model architecture and expanding the training data scale?

3.The improvements depicted in Figures 4, 5, and 6 (in the supplementary materials) do not appear to be particularly pronounced.

4.There are missing details regarding the training data used for the prompt learning framework.

5.The paper lacks references or calculation methods for NEFs.

**Questions:**

1.I understand that the authors propose the module's function is to bridge the gap between training and inference. However, I am curious why it is referred to as "prompt." It seems that the results produced by the model are not intended to serve as a "prompt" and do not interact with information from DMs, which may lead to reader confusion.

2.The readers will be a bit confused about Table 4. What do the numbers 10, 20, and 50 represent? What metrics are reflected in the results presented in the table? It is suggested to modify Table 4 or show more details more in caption.

3.Please provide detailed responses, as I will reassess the proposed methods and results based on your answers.

---

> ### Author Response · Authors · 2024-11-26
>
> Dear Reviewer TaPC,
>
> We greatly appreciate the tremendous effort you have put into carefully reviewing our paper!
>
> As the discussion period is nearly over, could you kindly take a moment to check our responses to address your concerns?
> We believe your comments have been incredibly helpful in improving our paper.
>
> Sincerely,
>
> Paper629 Authors

---

> > ### Comment · Reviewer_TaPC · 2024-11-26
> >
> > The authors have solved my problems. Based on the answers and experiments of the paper, i decide to raise my score from 5 to 6.

---

> ### Comment · Reviewer_TaPC · 2024-11-26
>
> The authors have solved my problems. Based on the answers and experiments of the paper, i decide to raise my score from 5 to 6.

---

### Official Review · Reviewer_xBN2 · 2024-11-03

**Soundness:** 3
**Presentation:** 4
**Contribution:** 3
**Rating:** 8
**Confidence:** 3

**Summary:**

This work reiterates the issue of exposure bias in diffusion models that is caused by score estimation and discretization errors. The work proposes a prompt prediction model, trained to simulate exposure bias, that can reduce the gap between the training trajectory and a typical sampling trajectory by introducing a reasonable overhead (5\%). The authors show experiment results on a few benchmark datasets (CIFAR-10, CelebA and ImageNet), demonstrating improvements in FID and IS.

**Strengths:**

The paper is well motivated and well articulated and easy to follow. The method makes intuitive sense and the experiments on a few benchmark datasets further support the authors' claims. The method can be easily implemented and integrated with various diffusion model architectures.

**Weaknesses:**

While the experiments focus on improving image quality, is there a possibility of trade-offs such as loss of diversity that could be caused by applying the prompt adjustment? I think additional experiments that take into account possible trade-offs could further strengthen the method.

**Questions:**

I have made a suggestion as part of my weaknesses statement. It would be interesting to knopw the authors' thoughts on whether diversity of generations could be affected.

---

### Meta-Review · Area_Chair_K24Y · 2024-12-19

**Metareview:**

This paper proposes a prompt prediction model to alleviate the training-sampling discrepancy in diffusion models.

The paper received unanimous accept recommendations (reviewer 3M1k's review is short and uninformative, so I largely disregard it). Most of the concerns that the reviewers raised have been addressed during the rebuttal stage.

The results presented in this paper could potentially be interesting to the general diffusion model community. I recommend accept.

**Additional Comments On Reviewer Discussion:**

Reviewers have been concerned that: this approach may result in diversity loss or other trade-offs; baselines are relatively simple and the improvements are not significant; the presentation can be improved, etc. Most of them have been addressed during rebuttal.

In the future, authors may consider apply this approach to text-to-image generation, as well as trying high-resolution generation.

---

### Decision · Program_Chairs · 2025-01-22

Accept (Spotlight)